# Biparametric vs. Multiparametric MRI in the Detection of Cancer in Transperineal Targeted-Biopsy-Proven Peripheral Prostate Cancer Lesions Classified as PI-RADS Score 3 or 3+1: The Added Value of ADC Quantification

**DOI:** 10.3390/diagnostics14151608

**Published:** 2024-07-25

**Authors:** Elena Bertelli, Michele Vizzi, Chiara Marzi, Sandro Pastacaldi, Alberto Cinelli, Martina Legato, Ron Ruzga, Federico Bardazzi, Vittoria Valoriani, Francesco Loverre, Francesco Impagliazzo, Diletta Cozzi, Samuele Nardoni, Davide Facchiano, Sergio Serni, Lorenzo Masieri, Andrea Minervini, Simone Agostini, Vittorio Miele

**Affiliations:** 1Department of Radiology, Careggi University Hospital, 50134 Florence, Italy; michele.vizzi@unifi.it (M.V.); sandropastacaldi87@gmail.com (S.P.); albycinelli@gmail.com (A.C.); martina.legato@gmail.com (M.L.); ron.ruzga@unifi.it (R.R.); federico.bardazzi@unifi.it (F.B.); vittoria.valoriani@gmail.com (V.V.); francesco.loverre@unifi.it (F.L.); francesco.impagliazzo@unifi.it (F.I.); dilettacozzi@gmail.com (D.C.); agostinis@aou-careggi.toscana.it (S.A.); vmiele@sirm.org (V.M.); 2Department of Statistics, Informatics and Applications “G. Parenti” (DiSIA), University of Florence, 50134 Florence, Italy; chiara.marzi@unifi.it; 3Unit of Urological Minimally Invasive, Robotic Surgery and Kidney Transplantation, Careggi Hospital, University of Florence, 50134 Florence, Italy; samuele.nardoni@unifi.it (S.N.); davidefacchiano@yahoo.it (D.F.); sergio.serni@unifi.it (S.S.); lorenzo.masieri@unifi.it (L.M.); 4Department of Experimental and Clinical Medicine, University of Florence, 50134 Florence, Italy; andrea.minervini@unifi.it; 5Unit of Oncologic Minimally-Invasive Urology and Andrology, Careggi Hospital, 50134 Florence, Italy

**Keywords:** prostate cancer, biparametric MRI, multiparametric MRI, prostate biopsy, ADC

## Abstract

Background: Biparametric MRI (bpMRI) has an important role in the diagnosis of prostate cancer (PCa), by reducing the cost and duration of the procedure and adverse reactions. We assess the additional benefit of the ADC map in detecting prostate cancer (PCa). Additionally, we examine whether the ADC value correlates with the presence of clinically significant tumors (csPCa). Methods: 104 peripheral lesions classified as PI-RADS v2.1 score 3 or 3+1 at the mpMRI underwent transperineal MRI/US fusion-guided targeted biopsy. Results: The lesions were classified as PI-RADS 3 or 3+1; at histopathology, 30 were adenocarcinomas, 21 of which were classified as csPCa. The ADC threshold that maximized the Youden index in order to predict the presence of a tumor was 1103 (95% CI (990, 1243)), with a sensitivity of 0.8 and a specificity of 0.59; both values were greater than those found using the contrast medium, which were 0.5 and 0.54, respectively. Similar results were also found with csPCa, where the optimal ADC threshold was 1096 (95% CI (988, 1096)), with a sensitivity of 0.86 and specificity of 0.59, compared to 0.49 and 0.59 observed in the mpMRI. Conclusions: Our study confirms the possible use of a quantitative parameter (ADC value) in the risk stratification of csPCa, by reducing the number of biopsies and, therefore, the number of unwarranted diagnoses of PCa and the risk of overtreatment.

## 1. Introduction

In recent years, prostate cancer incidence and mortality rates have been reduced or stabilized in many countries, with more pronounced declines in countries with high human development. These trends can be explained by a decline in PSA testing (incidence) and improvements in treatment (mortality) [1,2,3].

In Italy, PCa is the most frequent neoplasm in men (20% of diagnosed cancers) older than 50 years, with about 34.800 new cases in 2017. As for other neoplasms, there is a north–south gradient between different regions, from 153.5 cases per year/100.000 among residents in northern Italy, 139.3 cases per year/100.000 in the center (10%), and 109.5 cases per year/100.000 (29%) in the south [1].

Multiparametric magnetic resonance imaging (mpMRI) is an established tool for the management of patients with clinical suspicion of PCa. In fact, the latest European Association of Urology (EAU) guidelines [4] recommend mpMRI before prostate biopsy in biopsy-naïve patients and in patients with a prior negative biopsy. In the case of a positive mpMRI (i.e., PI-RADS ≥ 4), the EAU guidelines recommend combining a targeted biopsy with perilesional sampling. Instead, when the result of the mpMRI is indeterminate (PI-RADS = 3) and clinical suspicion of PCa is very low (PSA density < 0.10 ng/mL/cc, negative DRE findings, no family history), the latest EAU guidelines recommend omitting a biopsy and offering PSA monitoring; otherwise, a targeted biopsy with perilesional sampling should be considered.

The so-called ‘MRI pathway’ reduces the risk of over-diagnosis by half, at the cost of delaying detection of intermediate-risk cancers in a small percentage of patients [5,6,7]. 

Moreover, mpMRI, incorporating T2-weighted images (T2w), diffusion-weighted imaging (DWI) and dynamic contrast-enhanced (DCE) MRI sequences, increases PCa detection accuracy, and could guide the targeted biopsy to the suspicious lesions, whilst reducing the risk of unnecessary procedures or false negative outcomes [4].

In 2012, the American College of Radiology and the European Society of Urogenital Radiology introduced the Prostate Imaging and Reporting Data System (PI-RADS) to standardize image acquisition, interpretation, and reporting for multiparametric MRI (mpMRI). This system was subsequently updated with version 2 (v2) in 2015 and version 2.1 (v2.1) in 2019 [8,9].

According to the latest PI-RADS v2.1 guidelines [8], dynamic contrast-enhanced (DCE) imaging does not contribute to the evaluation of the transition zone (TZ) and is now only considered a “tie-breaker” for indeterminate lesions in the peripheral zone (PZ). Therefore, with high-quality T2w and DWI sequences, DCE MRI is often unnecessary [10].

Given the limited role of DCE-MRI, there is increasing interest in conducting prostate MRI without it, known as “biparametric MRI” (bpMRI). Omitting DCE-MRI offers several potential benefits, including reducing the time and cost of the MRI procedure and avoiding the adverse effects of contrast media. Several studies have demonstrated the effectiveness of bpMRI in detecting clinically significant prostate cancer (csPCa) in men who are biopsy-naïve, as well as those who have had a prior negative biopsy [8].

Despite DCE-MRI’s limited influence on the overall PI-RADS assessment category, evidence indicates that in some cases, DCE-MRI can aid in detecting csPCa in both the PZ and the TZ. In clinical practice, DCE-MRI is sometimes considered a “safety-net” or “backup” sequence, particularly when DWI and T2-weighted (T2W) images are compromised by artifacts or an insufficient signal-to-noise ratio (SNR). Additionally, DCE-MRI can be especially valuable for radiologists with less experience [11,12].

Conflicting results have been published regarding the added value of DCE imaging in detecting and characterizing TZ tumors. In recent studies, no enhancement characteristics could significantly differentiate malignant from benign lesions [9,13].

The poor performance of DCE imaging in detecting and characterizing TZ tumors is probably due to the fact that enhancement patterns between cancers and BPH nodules are similar, as suggested by prior studies [14].

According to the PI-RADS v2.1 recommendations, the scoring of peripheral zone lesions is based on the subjective interpretation of the signal intensity (SI) in high b-value DWI and the corresponding apparent diffusion coefficient (ADC) map. Multiple studies have demonstrated an inverse correlation between ADC values and the Gleason score, and a variety of methods for interpreting ADC values to diagnose prostate cancer and predict its aggressiveness have been used [15].

The ADC values calculated from DWI have been shown to correlate with PCa aggressiveness. Quantitative analysis of ADC values could prove to be a useful tool, as an additive variable in risk stratification for the diagnosis of csPCa. It could enable fewer biopsies and, consequently, reduce the unwanted diagnosis of non-clinically significant PCa [15].

We analyze the possible role of the mean ADC value, analyzed as a single parameter, in tumor detection, and, in particular, its possible role in the detection of clinically significant PCa.

Furthermore, we compared this datum (the mean ADC value) and the third parameter of multiparametric MRI, DCE imaging, in terms of sensitivity and specificity. Based on recent literature, this comparison is highly significant, as evaluating the ADC value might eliminate the need for a contrast medium, and instead lead to the employment of the simpler biparametric MRI rather than the multiparametric MRI.

Our work aims to compare a threshold of ADC values in a bpMRI study, with DCE imaging as a “tie-breaker sequence” in the mpMRI for the detection of PCa and, especially, for the detection of clinically significant PCa (Gleason score greater than or equal to 7) peripheral prostate lesions classified as PI-RADS 3 or 3+1. Specifically, we identify two main aims: (1) to compare the mpMRI evaluation in terms of the use of DCE imaging with bpMRI, in terms of threshold of ADC values, in the detection of PCa and (2) to find out whether the ADC value correlates not only with the presence of a tumor, but also with the presence of a clinically significant one.

## 2. Materials and Methods

Our single-center, retrospective, observational study included 104 peripheral lesions in 82 patients, classified as PI-RADS v2.1 score 3 or 3+1 at an mpMRI, performed at the Uro-Nephrology Unit of the Radiology Department at a tertiary care University Hospital, between June 2020 to December 2021. The study was conducted in accordance with the Declaration of Helsinki and approved by the local Institutional Review Board. All patients underwent free-hand transperineal MRI/US fusion-guided targeted biopsy (fhTFTB), performed at our Radiology Department. All the biopsies were performed by 2 radiologists, with more than 30 years of experience in urogenital pathology and more than 10 years of experience in prostate MRI imaging and free-hand transperineal MRI/US fusion-guided targeted biopsy. 

We included biopsy-naïve patients with clinical suspicion of PCa (total serum PSA > 3 ng/mL or PSA > 2 ng/mL and a positive family history or PSA density > 0.15 ng/mL/cc) or a suspicious digital rectal examination (DRE).

All patients who had already had a random or targeted biopsy in the previous 12 months were excluded, and we also excluded patients who underwent a fhTFTB in another institution.

### 2.1. MRI

All mpMRI were performed using a 1.5 T MR scanner, equipped with an anterior pelvic phased-array 18-channel coil and a posterior spine phased-array 32-channel coil (Magnetom Aera, Siemens Medical Systems, Erlangen, Germany) at the Uro-Nephrology Unit of our Radiology Department.

According to PI-RADS v2.1, our acquisition protocol included high-resolution: T2-weighted turbo spin-echo (TSE) sequences in the axial, sagittal, and coronal planes (slice thickness 3 mm without gap; matrix 272 (P) × 320 (F); FOV (200mm × 200 mm); T1-weighted pre-contrast spin-echo (SE) sequences in the axial plane; multi b-value DWI (50, 500, 800, 1000 s/mm^2^) (EPI-DWI) sequences from which corresponding ADC maps were obtained; multi b-value DWI (1400-1800 s/mm^2^) (EPI-DWI) sequences; and DCE imaging assessments with fat suppression gradient-echo 3D T1W sequences, with high time resolution (<7 s) and time intensity curve evaluation.

The detailed technical parameters of our protocols are described in the Supplementary Materials to our previous work [16].

#### 2.1.1. MRI Analysis

A total of 163 mpMRI exams were interpreted by radiologists specialized in urogenital pathology (5 to 15 years of experience in prostatic MRI), who assigned a score of 1–5 for T2WI, a score of 1–5 for DWI, and positive or negative score for DCE-MRI, according to PI-RADS v2.1, and determined the overall PI-RADS v2.1 assessment category for the PZ and the TZ.

The lesions in the peripheral zone with a DWI score of 3 were upgraded to a PI-RADS score of 4 based on the positivity of the DCE-MRI, defined as PI-RADS “3+1” in order to distinguish them from the “DWI 4” PI-RADS 4 score. 

In the 163 mpMRI exams, 247 index lesions were identified: 76 lesions of which were classified as PI-RADS v2.1 3, 142 lesions were PI-RADS v2.1 4 of which 49 were “3+1”, and 29 lesions were PI-RADS v2.1 5.

Among them, we selected only the peripheral lesions. Hence, 104 lesions were classified as PI-RADS v2.1 3 (55 lesions) or 3+1 (49 lesions) in the peripheral zone and, subsequently, were interpreted without DCE-MRI by 2 readers from the urogenital radiologist group, blinded to the biopsy and the previous mpMRI results. 

#### 2.1.2. Quantitative Analysis

Quantitative analysis of the ADC values was conducted retrospectively by two readers, blinded to the biopsy results, using Syngo.via (Syngo.via VB10B, MM Oncology workflow, Siemens Medical Systems, Erlangen, Germany). They centered on a ROI (region of interest) on the PI-RADS v2.1 3 or 3+1 biopsied lesion identified on the ADC maps, including in the ROI involving at least 50% of the lesion and excluding margins. 

### 2.2. Fusion Targeted Biopsy

All the lesions were considered suspicious for PCa and were biopsied.

All patients underwent a free-hand transperineal MRI/US fusion-guided targeted biopsy, using a virtual navigation platform (MyLabTMTwice Esaote, Genoa, Italy).

A TRUS was performed using sagittal views, with a biplane endorectal probe (5–13 MHz).

The biopsies were performed with a 16 G automatic tru-cut needle, by two radiologists with at least 25 years of experience in transperineal prostate biopsy, skilled in the virtual navigation procedure, and with, respectively, 10 and 5 years of experience in interpreting mpMRI results.

All suspicious mpMRI lesions were sampled. A total of 3–5 targeted cores for each lesion were obtained, in relation to their size. All patients also underwent 8–14 randomized standard prostate biopsy cores (6–10 samples for the peripheral zone and 2–4 samples for the transition zone) in relation to the prostate volume.

All patients were properly informed about the potential risks of the examination, and gave their written informed consent to the procedure.

### 2.3. Histopathological Analysis

The biopsy specimen analysis was performed by expert pathologists dedicated to urogenital pathology, working for the Histopathological Department at our University Hospital.

The reports included: the type of carcinoma, primary and secondary GS (per biopsy site and global), the percentage of high-grade carcinoma (global), and the linear extent of the carcinoma in mm [4,17]. PCa with a GS ≥ 3+4 or a CCL ≥ 5 mm was considered clinically significant. 

### 2.4. Statistical Analysis

Firstly, we evaluated the capability of the DCE-MRI protocol (so the mpMRI) to discriminate positive (malignant, PCa) from negative (benign) lesions (task #1), and clinically significant lesions from non-clinically significant ones (task #2). Specifically, we compared the PI-RADS v2.1 values with the related biopsy-based labels, computing the contingency matrix, the sensitivity, and the specificity.

Similarly, we evaluated the capability of the ADC maps to perform both tasks #1 and #2. In this case, as the average ROI ADC values were continuous and the optimal cut-off value for the lesion classification was not known a priori, we constructed the receiver operating characteristic (ROC) curve by varying the cut-off value [18]. Since lower ADC values generally correspond to a higher probability of malignant or clinically significant lesions, in constructing the ROC curve by varying the threshold, all lesions with a mean ADC in an ROI smaller than the threshold were classified as “positive” or “clinically significant”. The area under the ROC curve (AUROC) was computed for each task. We then identified the optimal operating point on the ROC curve, defined as the point with the highest Youden index (J = sensitivity + specificity − 1) [19]. This optimal operating point allowed us to define a threshold value for the ADC and compute the sensitivity and specificity. We performed the statistical analyses using STATA v17 and utilized in-house Python programs (employing numpy v1.21.5, matplotlib v3.4.3, scikit-learn v1.0.2, and pandas v1.4.0).

## 3. Results

We found 104 peripheral prostate lesions classified as PI-RADS v2.1 score 3 (55 lesions, Figure 1) or 3+1 (49 lesions) in 82 patients (mean age 64.6 years, range 44–78 years). The detection rate for PI-RADS 3 lesions at our institution was 11%.

The mean diameter of the lesion was 8.5 mm (range 4–30 mm). 

Histopathologic examination of the tissue samples derived from the free-hand transperineal MRI/US fusion-guided targeted biopsy revealed 30 adenocarcinomas (15 PI-RADS 3 and 15 PI-RADS 3+1), of which 21 classified as clinically significant tumors (11 PI-RADS 3 and 10 PI-RADS 3+1).

In particular, 9 (30%) lesions were histopathologically classified as Gleason score 3+3, 14 (46.7%) as Gleason score 3+4, and 7 (23.3%) as Gleason score 4+3 (Table 1 and Table 2). The mean value of the ADC was 1125 (interquartile range 1021.5–1245) for all the lesions, 1032.5 (interquartile range 97.8–1094) for the lesions classified as tumors, and 1020.9 (interquartile range 972–1082) for the tumors classified as clinically significant. As illustrated in Figure 1 and Figure 2 PI-RADS 3 and 3+1 lesions may appear only moderately hypointense in the T2-weighted sequence and exhibit a mild diffusion restriction on DWI, but are often hypointense in terms of the ADC. Therefore, it is important to determine the average ADC value, as shown in these figures. Specifically, in Figure 1, the mean ADC value is 1213, and histopathology revealed the tumor to be a GS 3+3. In Figure 2, the average ADC value is 1024, and the tumor was determined to be a clinically significant GS 4+3.

We tried to understand whether, in our group, the mean ADC value could predict the possibility of finding a tumor and, in particular, a clinically significant one, and compared the results with the sensitivity and specificity of the use of the contrast medium to understand whether the biparametric evaluation was comparable to the multiparametric one.

The multiparametric evaluation achieved a sensitivity equal to 0.50 and a specificity of 0.54 in detecting positive lesions (contingency matrix in Table 3), while it achieved a sensitivity equal to 0.48 and a specificity of 0.53 (contingency matrix in Table 4) in defining clinically significant lesions.

The ADC threshold that maximized the Youden index in order to predict the presence of a tumor was 1103 (non-parametric bootstrap 95% confidence interval (CI) of (990, 1243)), with a sensitivity of 0.8 and a specificity of 0.59; both values were greater than those found using a contrast medium, which were 0.5 and 0.54, respectively (Table 5, Figure 2, Figure 3 and Figure 4).

Similar results were also seen with clinically significant tumors, where the optimal ADC threshold was 1096 (95% CI (988, 1096)), with a sensitivity of 0.86 and a specificity of 0.59, compared to 0.48 and 0.53 for those observed with the use of a contrast medium (Figure 2).

## 4. Discussion

Several retrospective studies have demonstrated that an abbreviated bpMRI protocol without DCE-MRI allows the detection of csPCa, with diagnostic accuracy equivalent to that of a conventional, full mpMRI [11,12].

In a meta-analysis conducted by Niu et al. [20], bpMRI showed high sensitivity (81%) and also high specificity (77%) in detecting PCa. 

Choi et al. conclude that bpMRI is useful in stratifying the probability of csPCa [21]. 

Most research papers have concluded that bpMRI could replace mpMRI in the diagnosis of clinically significant PCa, with nearly the same accuracy in the detection of suspicious lesions. These lesions graded as PI-RADS 3 and 4 require, in any case, a DCE sequence. The use of a contrast medium is required, because a PI-RADS 3 lesion indicates an equivocal lesion, with a probability of developing clinically significant prostate cancer, whereas a PI-RADS 4 lesion indicates that cancer is more likely to be present [22].

In a recent review by Iacob et al. [22], which scanned 31 relevant articles regarding the advantages and disadvantages of bpMRI and mpMRI, the authors conclude that bpMRI can mostly be used for the detection of PCa, but cannot replace mpMRI in the other phases of PCa management.

Pan et al. [23], in a multivariate analysis that included 571 men who underwent bpMRI or mpMRI, found that the detection rate of PCa and csPCa by the two MRI techniques were comparable, with no statistically significant differences.

A recent review by Caglic [13] analyzed the role of bpMRI in the active surveillance setting, highlighting the pros and cons. The possible disadvantages are that DCE-MRI can act as a “safety sequence” for image quality in the context of poor-quality images, can detect new lesions, and is accurate in terms of staging (e.g., seminal vesicles invasion). 

One of the main advantages of bpMRI is the higher specificity compared to mpMRI for the detection of PCa. This is due to the relative non-specificity of DCE-MRI: as tumors, both prostatitis and highly vascularized benign prostate hyperplasia nodules, are characterized by rapid and significant enhancement. The authors conclude that bpMRI could be a reasonable approach in the setting of active surveillance, especially in combination with recent artificial intelligence solutions, which could mitigate the potential disadvantages [16,24,25,26]. 

In the literature, many authors have investigated the potential role of ADC measurements in improving the diagnosis of prostate cancer [27,28,29]. However, numerous studies do not report a threshold value, and more recent research focuses on the use of radiomics or deep learning for ADC map analysis [28,29]. Furthermore, some studies focus only on the transitional zone [30]. Only a few older studies (which therefore often use PI-RADS v2) examine the reliability of a threshold ADC value as an additional parameter for prostate cancer diagnosis. Jordan et al. [27] explore the role of ADC values and ADC categories (<800 and >800) in predicting csPCa. The authors conclude that both ADC values and categories help in diagnosing csPCa, but only in lesions with a PI-RADS score of 4.

A recent study by Tavakoli et al. investigates whether DCE-MRI in combination with ADC improves the prediction of csPCa (although the article is recent, the authors use PI-RADS v2) [31]. They conclude that DCE-MRI is not relevant in PI-RADS 3 lesion risk stratification, instead ADC is important in upgrading PI-RADS 3 and 4 lesions. Furthermore, the authors suggest including quantitative ADC measurements in a future version of PI-RADS.

Moreover, there is still no consensus in the literature on incorporating additional quantitative parameters, including clinical ones like PSA, PSA density, and PSA velocity, in the risk stratification of prostate cancer [32,33]. Therefore, this remains a potentially interesting area for further research.

The main limitations of this study are as follows:The number of patients is relatively small, although it should be noted that the cohort consists of a highly selected group of peripheral neoplastic lesions, all rated as PI-RADS 3 and 3+1, and all the lesions were histologically confirmed by transperineal MRI/US fusion-guided targeted biopsy;The variability of the ADC value differs between different MRI scanners, with the need to identify different suspected ADC value thresholds for each scanner. Scanner-specific ADC value thresholds should be identified in prostate MRI centers with an adequate caseload.

## 5. Conclusions

Our study confirms the possible use of a quantitative parameter, such as the ADC value, in the risk stratification of csPCa. This approach can reduce the number of biopsies and, consequently, the number of unnecessary prostate cancer diagnoses, thereby lowering the risk of overtreatment for these patients.

Looking ahead, future perspectives could include the introduction of additional quantitative parameters, including clinical ones, such as PSA, PSA density, and PSA velocity, in the risk stratification of prostate cancer. This would enhance the detection rate of bpMRI, further reducing the number of biopsies. Additionally, the implementation of artificial intelligence solutions holds promise in this field.

## Figures and Tables

**Figure 1 diagnostics-14-01608-f001:**
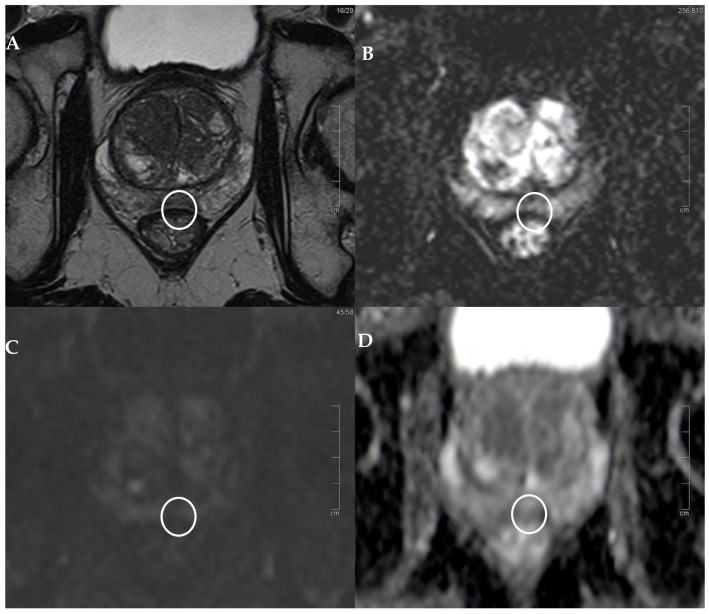
PI-RADS 3 lesion. (**A**) T2w sequence showing a moderate, hypointense, small (6 mm) lesion in left PZ pm middle gland. (**B**) The correspondent DCE-MRI sequence showing slight enhancement of the lesion (white circle). (**C**) DWI is slightly restricted, but the lesion (underlined by the white circle) is hypointense in terms of the ADC, (**D**) with an average ADC value of 1213. At histopathology: GS 3+3 lesion.

**Figure 2 diagnostics-14-01608-f002:**
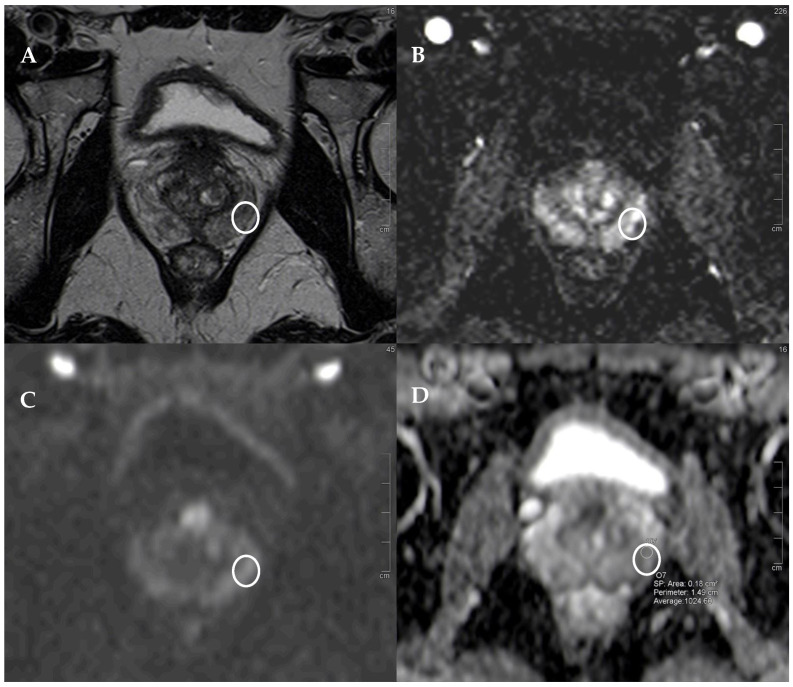
PI-RADS 3+1 lesion. (**A**) T2w sequence showing a moderate, hypointense, small (5 mm) lesion in left PZ pl middle gland, underlined by the white circle in all the images. (**B**) The correspondent DCE-MRI sequence showing significant enhancement of the lesions. The peripheral zone exhibits contrast enhancement in other areas that are bilateral and symmetrical, though less intense than the tumor area, indicating inflammatory phenomena (histopathologically confirmed). (**C**) DWI is slightly restricted, but the lesion is hypointense in terms of the ADC (**D**), with an average ADC value of 1024. At histopathology: GS 4+3 lesion.

**Figure 3 diagnostics-14-01608-f003:**
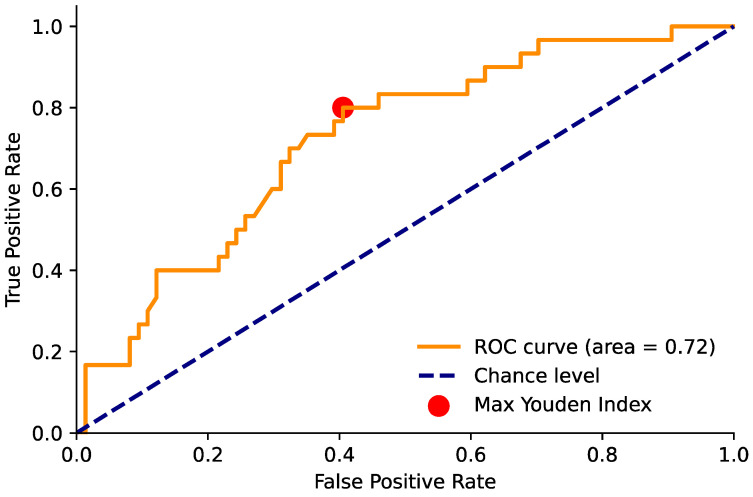
Area under the ROC curve for the ADC value and positivity for PCa. The red point indicates the maximum Youden index value (sensitivity = 0.80 and specificity = 0.59) and corresponds to an ADC threshold of 1103.

**Figure 4 diagnostics-14-01608-f004:**
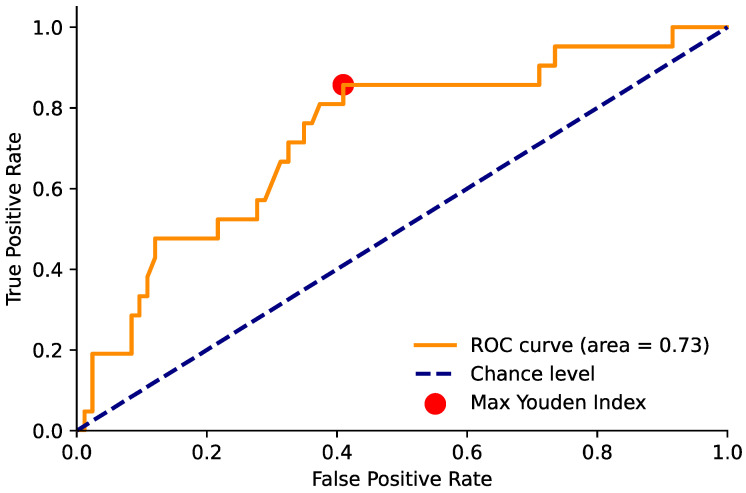
Area under the ROC curve for the ADC value and positivity for csPCa. The red point indicates the maximum Youden index value (sensitivity = 0.86 and specificity = 0.59) and corresponds to an ADC threshold of 1098.

**Table 1 diagnostics-14-01608-t001:** Characteristics of the study population.

Variable	Value
**Mean age, y**	64.6 (44–78)
**Mean diameter, mm**	8.5 (4–30)
**Total lesions**	104
** *Adenocarcinomas* **	49 (57.1%)
** *Clinically significant* **	21 (20.2%)
**PI-RADS**	
*3*	55 (52.9%)
*3+1*	30 (28.8%)
**Gleason score**	
*3+3*	9 (30%)
*3+4*	14 (46.7%)
*4+3*	7 (23.3%)

**Table 2 diagnostics-14-01608-t002:** Lesion characteristics.

	104 Lesions	30 Adenocarcinomas	21 Clinically Significant
DCE−	55	15	11
DCE+	49	15	10
ADC above threshold	50 (1103)	6 (1103)	3 (1103)
ADC under threshold	54 (1103)	24 (1103)	18 (1103)

**Table 3 diagnostics-14-01608-t003:** Contingency matrix for task #1 using DCE-MRI technique.

	Positivity	
PI-RADS v2.1	0	1
3	40	15
3+1	34	15

**Table 4 diagnostics-14-01608-t004:** Contingency matrix for task #2 using DCE-MRI technique.

	Clinically Significant	
PI-RADS v2.1	0	1
3	44	11
3+1	39	10

**Table 5 diagnostics-14-01608-t005:** Summary of ROC analysis results. We defined the estimate of the ADC cut-offs according to the maximal Youden index.

	DCE-MRI and Positivity for PCa	DCE-MRI and Positivity for Clinically Significant PCa	ADC and Positivity for PCa	ADC and Positivity for Clinically Significant PCa
AUROC	-	-	0.72	0.73
ADC cut-offs	-	-	1103	1098
Sensitivity	0.50	0.48	0.80	0.86
Specificity	0.54	0.53	0.59	0.59

## Data Availability

The data presented in this study are available on request from the corresponding author due to privacy reasons.

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
