# Peer review of "Biparametric vs. Multiparametric MRI in the Detection of Cancer in Transperineal Targeted-Biopsy-Proven Peripheral Prostate Cancer Lesions Classified as PI-RADS Score 3 or 3+1: The Added Value of ADC Quantification"

_diagnostics, 2024, doi:10.3390/diagnostics14151608_

Round 1

Reviewer 1 Report

Comments and Suggestions for Authors

General impression:

Overall, the manuscript presents an interesting and relevant idea, addressing the increasing issue of imperfections in the diagnostic pathway of prostate cancer (PCa). The primary focus is on using ADC value measurements to predict clinically significant PCa in suspicious lesions, a topic well-explored in the literature. Such studies contribute to our understanding of current knowledge in prostate cancer diagnosis, which translates into clinical practice.

It is noteworthy that the manuscript is engaging, and despite the complexity of the topic, it remains understandable to the average reader. According to current conventions, the results of a study should be described in the past tense once the study has been conducted. Additionally, there are several language errors that make it difficult to understand, suggesting that the authors should consider seeking assistance from an English editor.

Abstract:

Please provide a clear explanation of the study's aim instead of the sentences "We evaluate the added value of the ADC map for detecting PCa. Furthermore, we investigate whether the ADC value correlates with the presence of a clinically significant tumor (csPCa).”

Line 24: "a sensibility of 0.8" - I'm quite sure the author meant sensitivity, and this error repeats throughout the entire manuscript.

Apart from that, the abstract is rather concise and informative.

Introduction:

The authors stray a bit somewhat from the main topic in the introduction. There is no need to elaborate on the general epidemiology of prostate cancer, given that the article is aimed at a professional journal with readers who are specialists in the field. The section discussing the epidemiological situation in Italy was more compelling.

Regarding that, I would suggest rephrasing the first four paragraphs of the already lengthy introduction and rewriting it. Attention should be drawn to the growing problem of clinical decision making in PI-RADS 3 cases, as numerous different approaches were tested in such setting.

Lines 51-56: Please review this statement in light of the current edition of EAU guidelines. The recommended treatment has recently been updated for PI-RADS 3 cases.

(i.e. PIRADS ≥ 3) - a dash is missing

The subject of performing biparametric MRI alone, the reasons for this approach, and the challenges it entails are clearly stated in the remainder of the introductory paragraph. The section discussing indications where mpMRI is preferred over bpMRI is very informative. The aims should be separated into primary and secondary for clarity.

Methods and Results:

Line 127: monocentric - a single center study

Line 129: „a tertiary care University Hospital 129 between June 2020 to December 2021” - Why is the cohort from three years ago? Perhaps it would be worthwhile to expand it with new cases that have undoubtedly been referred to the highest-level referral center.

I would refrain from describing every lesion as PI-RADS v2.1 because it is completely unintelligible.

I believe PI-RADS is sufficient.

The detection rate of PI-RADS 3 lesions is considered an indicator of MRI report quality and should not exceed 15% of all detected PI-RADS ≥3 cases. Please describe the detection rate of PI-RADS 3 lesions in your study.

Lines 238-243: These considerations should be described in the introduction.

Lines 252-253: Misuse of „sensibility” once again.

For all constructed contingency tables, appropriate statistical tests with p-values should be provided.

Figure 2. The calculated threshold of ADC should be indicated on the graph.

Discussion

The study revolves around calculating the ADC values that could discriminate between the presence or absence of csPCa.

In the discussion, insufficient space is devoted to other publications that also evaluate the ADC cutoff point in detecting csPCa, which is the main topic of the manuscript. In my opinion, the article is an interesting study of the problem. 

However, it lacks linguistic correctness, is conducted on a very small number of patients for a cohort from four years ago, and issues regarding ADC have not been sufficiently discussed. In light of these concerns, I am afraid the manuscript needs thorough restructuring before resubmission.

Comments on the Quality of English Language

Extensive editing of English language required

Author Response

Please find attached our response to Reviewer 1 (highlighted in red within the text).

Reviewer 2 Report

Comments and Suggestions for Authors

I have read the manuscript titled "Biparametric vs Multiparametric MRI in the detection of cancer in transperineal targeted-biopsy proven peripheral prostate cancer lesions classified as PIRADS score 3 or 3+1: the added value of ADC quantification." The authors address a highly interesting topic regarding prostate cancer detection from both radiological and urological perspectives. The objective of this study is commendable, and I believe it therefore warrants further investigation. In general, I believe the manuscript is well-written and well-designed; however, I have identified several issues that require attention. I suggest that a revision is necessary before this paper is ready for publication.

1. Please check the abbreviations used throughout the manuscript. For instance, line 124: "Pca" should be changed to "PCa," and line 275: "clinically significant prostate cancer" should be abbreviated as "csPCa." Please carefully review the entire manuscript to rectify similar issues.

2. Some minor English editing is required. For instance, line 54-55: "the EAU Guidelines recommend to combine targeted ....., and to perform targeted..." should be changed to "the EAU Guidelines recommend combining targeted..., and performing..." Please also be cautious with punctuation and flow throughout the manuscript.

3. Please consider including a flowchart to reflect the detailed recruitment process.

4. In the subchapter titled "2.1. MRI," the authors may wish to include a table detailing the technical parameters of the mpMRI sequences, such as repetition time, echo time, slice thickness, etc.

5. Please clarify the annotations in Figures 1 and 4: "(D) with an average of 1213," and "(D) with an average of 1024," respectively.

6. For clarity, in the subchapter titled "2.4. Statistical Analysis," the authors are encouraged to indicate the methods used for the statistical analysis. Were R Studio, Python, or any other software tools used for processing?

7. As the Youden index is calculated based on sensitivity and specificity, the authors may consider including the equation for calculating the Youden index.

8. Please include table notes to indicate the measurements used in the tables. For instance, in Table 5, note that "sens" stands for sensitivity and "spec" for specificity.

9. The discussion and conclusion sections are not solid enough and have room for optimization. For instance, the authors may consider elaborating a table for recent research using bpMRI in the detection of both non-clinically significant PCa and csPCa, demonstrating sample size, sensitivity, specificity, accuracy, etc., for more precise evidence support. Additionally, the authors may discuss the diagnostic value of using 1.5 T versus 3.0 T mpMRI machines. Moreover, the authors mentioned the potential use of artificial intelligence (AI) in assisting PCa diagnosis. This could be expanded to provide more balanced and advanced viewpoints. Recent deep learning models have gained significant attention in terms of PCa detection, segmentation, and diagnosis. The future need for explainable AI is also anticipated. The authors are encouraged to assess relevant literature and improve the strength of their arguments.

Comments on the Quality of English Language

English editing is required, aiming at reducing grammatical errors and improving flow and clarity.

Author Response

Please find attached our response to Reviewer 2 (highlighted in red within the text).

Round 2

Reviewer 2 Report

Comments and Suggestions for Authors

I have checked the revisions the authors made and found that they have extensively modified their original submission. The authors have addressed the majority of my previous concerns, and I therefore recommend acceptance of the manuscript in its current form.

Author Response

We thank the Reviewer for the valuable comment.